# Understanding and Quantifying Reliability in Object Detection Transformers

## Abstract

Object detection is a computer vision task with significant utility, with real-world applications ranging from autonomous driving to warehousing and medical image analysis. Recently, Object Detection Transformers (DETR) have emerged as a prominent approach, offering an end-to-end prediction pipeline. The core innovation of DETR lies in the introduction of object queries, which attend to each other throughout the Transformer decoder layers and provide a set of outputs (i.e., bounding boxes and class probabilities) for a given image. Despite these advances, the mechanisms behind how these predictions are generated and interact are not well understood. For this reason, this paper explores the underlying dynamics of DETR's predictions and presents empirical findings that highlight how different predictions within the same image serve distinct roles, leading to varying levels of reliability across those predictions. In particular, we investigate the significance of differentiating between positive and negative predictions for uncertainty quantification (UQ) in DETR. Leveraging these insights, we propose novel post hoc UQ methods to quantify the image-level reliability of DETR and demonstrate their effectiveness through numerical analysis.

## 1 Introduction

Object detection is an essential task in computer vision, with applications that span various domains, including autonomous driving, warehousing, and medical image analysis. Traditional approaches to object detection have relied on Convolutional Neural Networks (CNNs) [Girshick et al., 2014, Ren et al., 2015, Redmon et al., 2016, He et al., 2017] to identify and locate objects within images. However, the recent introduction of Object Detection Transformers (DETR) [Carion et al., 2020] has revolutionized the field by offering an end-to-end prediction pipeline, where the model directly predicts a set of bounding boxes and class probabilities for each object in an image.

The core innovation of DETR lies in the use of a Transformer encoder-decoder architecture (see Section 2 for more detail), enabling the model to generate predictions in an end-to-end manner and enhancing scalability. This paradigm shift has led to the exploration of various DETR variants — such as Deformable-DETR [Zhu et al., 2020], DINO [Zhang et al., 2022], and Cal-DETR [Munir et al., 2024] — positioning them as potential foundation models for object detection tasks. Despite these advancements, the inner workings of *how **set predictions** are generated and interact within the Transformer decoder layers* remain poorly understood.

This paper examines this issue from the perspective of reliability/uncertainty quantification of DETR's[1] predictions. Specifically, we aim to answer **two key research questions**: (1) Do all

---

[1]In this paper, DETR refers broadly to the original model and its variants.

Submitted to Workshop on Bayesian Decision-making and Uncertainty, 38th Conference on Neural Information Processing Systems (BDU at NeurIPS 2024). Do not distribute.

predictions behave similarly and exhibit a consistent correlation with the model's reliability? (2) If not, how can we accurately assess DETR's reliability for a given input image?

Our preliminary findings indicate that different predictions from DETR in the same image play distinct roles resulting in varying levels of reliability. Specifically, as the decoder processes each object query through multiple layers, interactions among predictions occur, leading to the refinement of one positive query per object while others (i.e., negatives) provide support (Section 4.1).

To this end, this paper investigates the importance of distinguishing between these positive and negative predictions from the perspective of post hoc UQ in DETR, which is one of our key contributions (Section 4.2). We empirically observe that the confidence of negative queries inversely correlates with reliability, in contrast to positive queries, where higher confidence typically indicates greater reliability. Based on these insights, we propose novel methods to more accurately quantify the image-level reliability of the DETR model and conduct experiments evaluating them (Section 4.3).

## 2 Background: Object Detection Transformers (DETR)

The structure of DETR can be broadly divided into two main components: the Transformer encoder, which extracts a collection of features from the given image; and the Transformer decoder, which uses these features to make predictions. We refer to Figure 2 in the Appendix for an illustration.

In addition to the features extracted by the encoder, the decoder's input consists of $N$ (typically several hundreds) learnable embeddings, also known as *object queries*. Each decoder layer is composed of a self-attention module among object queries and a cross-attention module between each object query and the features. After processing the queries through several decoder layers, the model produces the $N$ final representation vectors that are converted into bounding boxes and class labels via a shared feedforward network, $f_\phi$. Together, these predictions form the final outputs, making DETR's predictions essentially an $N$-element set.

The encoder follows the common structure of standard computer vision models and is based on pre-trained models, whose reliability has been widely explored [Shelmanov et al., 2021, Sharma et al., 2021, Vazhentsev et al., 2022, Park et al., 2023]. This foundation further enables the use of prominent post hoc uncertainty quantification (UQ) techniques, such as Monte Carlo dropout [Gal and Ghahramani, 2016] and distance-based out-of-distribution (OOD) detection methods [Lee et al., 2018, Tack et al., 2020].

However, despite the decoder being the predominant component for object detection, there is a gap in understanding and quantifying its reliability due to its unique structural characteristics: set prediction. Therefore, this paper delves into the roles and behavior of these predictions and presents a methodology to estimate the reliability of the decoder in DETR for object detection tasks.

## 3 Problem Statement: Quantifying the Image-Level Reliability

We first introduce a formal definition of *image-level reliability* by examining the model's overall object detection performance on the image.

**Definition 1.** Let $x^*$ be a test image, and $\mathcal{T}_{x^*}$ denote the set of ground truth objects in the image. The outputs of the DETR, parameterized by $\theta$, for the image are represented by $\widetilde{\mathcal{T}}_\theta(x^*)$. Each object is represented by a bounding box and a class label (with probability). We define image-level reliability as a measure of how accurately and confidently the predictions match the ground truth objects:

$$\mathsf{ImReli}(x^*; \theta) \triangleq \mathsf{Acc}\big(\widetilde{\mathcal{T}}_\theta(x^*),\ \mathcal{T}_{x^*}\big). \tag{1}$$

where we quantify Acc using standard performance metrics such as precision, recall, and negative DETR matching cost (i.e., neg. MC). Details on how these metrics are computed can be found in Lin et al. [2014] and Carion et al. [2020].

To the best of our knowledge, existing UQ techniques focus mainly on object-level analysis (e.g., [Du et al., 2022a,b, Wilson et al., 2023, Sbeyti et al., 2024]) which are often conducted on predefined ground truth objects. Once the ground truth objects are provided, the bipartite matching algorithm (detailed further in Appendix A) can be used to find the best matching prediction for each object based on the alignment of the class label and bounding box. In this paper, we refer to these queries as *positive queries*, while the remaining queries are referred to as *negative queries*.

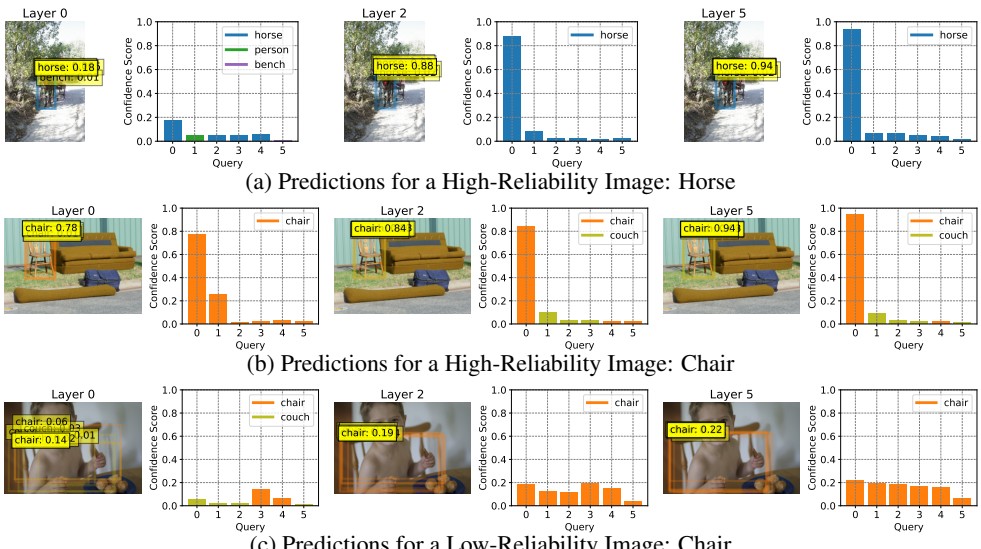

(a) Predictions for a High-Reliability Image: Horse

(b) Predictions for a High-Reliability Image: Chair

(c) Predictions for a Low-Reliability Image: Chair

Figure 1: Predictions made by Cal-DETR for high- and low-reliability images. The positive query (0) and the five negative queries (1-5) with the largest intersection of union (IoU) are presented. The model refines its predictions through each decoder layer, culminating in a high-confidence positive query while neighboring negative queries remain less confident.

In real-world scenarios, however, ground truth annotation is unavailable (i.e., the ground truth positive queries are unknown). Furthermore, predictions far outnumber ground truth objects, leaving it unclear whether reliability should be assessed for all predictions, or a subset (and, if so, which subset?). Thus, extending these methods to DETR raises a non-trivial question, which this paper aims to address.

## 4 Proposed Methods & Empirical Findings

### 4.1 Inside the Black Box: Exploring the Anatomy of DETR's Predictions

In developing a suitable UQ method, we begin by examining DETR's prediction process. Since the Transformer decoder outputs only representation vectors, investigating their evolution across layers is not straightforward. We address this by reapplying the final feedforward network that operates on the last layer, $f_\phi$, to the intermediate layers. This enables the transformation of each representation vector into its associated bounding box and class label. This is feasible due to the alignment of intermediate representations, facilitated by residual connections between decoder layers [Chuang et al., 2023, Munir et al., 2024]. Sample visualizations are in Figures 1, 3, and 4.

In the first decoder layer, the model appears to explore the encoded image features, producing varied queries that result in various plausible predictions. In this early stage, the distinction between positive and negative queries can be ambiguous (e.g., Figure 1a). However, the self-attentions through the subsequent decoder layers progressively refine these predictions. By the final layer, the model selects a single query (i.e., positive) and assigns a confidence score based on its understanding on the image and the object. In contrast, the confidence scores for neighboring queries (i.e., negatives) do not increase to the same extent as the positives and even decrease (e.g., Figure 1b) for reliable images. Hence, for a *reliable* image, we observe a **large gap between the positive and negative queries**.

Moreover, having queries with low confidence scores does not necessarily imply low reliability; in fact, empirical observations show that high image-level reliability actually correlates with low confidence scores in negative queries (Note that the correlation for $Conf^-$ is negative, as shown in Table 1.). This underscores the importance of accurately distinguishing the positive query within the entire set to achieve accurate UQ in DETR.

Another notable observation is that, for *unreliable* image (e.g., Figure 1c), the confidence score of the positive query does not grow significantly, unlike in reliable cases. In contrast, the confidence of negative queries increase, resulting in a **small gap between the positive and negatives**.

Table 1: Comparison of the proposed method (ContrastiveConf) with baseline methods on their Kendall's $\tau$ correlation coefficient with the image-level reliability in Equation 1, with Cal-DETR. Note that we use the "negative(-)" $\mathsf{Conf}^-$ due to the inverse correlation between average confidence of negative queries and reliability, as discussed in Section 4.1. The GT (oracle), highest, and second-highest scores are highlighted in light gray, blue and green, respectively.

| Method | Separation | COCO (in-distribution) | | | Cityscapes (near OOD) | | | Foggy Cityscapes (OOD) | | |
|---|---|---|---|---|---|---|---|---|---|---|
| | | Precision | Recall | neg. MC | Precision | Recall | neg. MC | Precision | Recall | neg. MC |
| $\mathsf{Conf}^+$ | Entire | -0.412 | -0.424 | -0.380 | -0.221 | -0.223 | -0.263 | -0.127 | -0.130 | -0.108 |
| $\mathsf{Conf}^+$ | Thr | 0.369 | 0.305 | 0.623 | 0.247 | 0.214 | 0.393 | 0.220 | 0.181 | 0.311 |
| | Top-$k$ | 0.073 | -0.011 | -0.380 | 0.064 | 0.036 | -0.263 | 0.078 | 0.047 | -0.108 |
| | NMS | -0.401 | -0.419 | -0.362 | -0.210 | -0.214 | -0.246 | -0.118 | -0.123 | -0.097 |
| | GT | 0.461 | 0.361 | 0.808 | 0.449 | 0.447 | 0.795 | 0.466 | 0.456 | 0.822 |
| $-\mathsf{Conf}^-$ | Thr | 0.422 | 0.416 | 0.452 | 0.273 | 0.267 | 0.337 | 0.194 | 0.185 | 0.185 |
| | Top-$k$ | 0.294 | 0.273 | 0.429 | 0.171 | 0.185 | 0.298 | 0.129 | 0.131 | 0.155 |
| | NMS | 0.409 | 0.395 | 0.427 | 0.224 | 0.212 | 0.292 | 0.140 | 0.132 | 0.132 |
| | GT | 0.417 | 0.405 | 0.439 | 0.214 | 0.205 | 0.235 | 0.146 | 0.140 | 0.122 |
| ContrastiveConf | Thr | 0.423 | 0.386 | 0.610 | 0.304 | 0.280 | 0.449 | 0.252 | 0.219 | 0.303 |
| | Top-$k$ | 0.368 | 0.311 | -0.110 | 0.251 | 0.230 | -0.067 | 0.193 | 0.173 | 0.031 |
| | NMS | 0.052 | -0.013 | 0.174 | 0.010 | -0.014 | 0.072 | 0.026 | 0.008 | 0.067 |
| | GT | 0.481 | 0.385 | 0.810 | 0.458 | 0.451 | 0.784 | 0.472 | 0.459 | 0.813 |

## 4.2 Proposed Method: Quantifying Reliability in DETR

Based on the aforementioned observations, we propose a novel post hoc UQ approach by contrasting the confidence score (i.e., reliability) of positives and negatives:

$$\mathsf{ContrastiveConf}(\boldsymbol{x}^*) = \mathsf{Conf}^+(\boldsymbol{x}^*) - \lambda \mathsf{Conf}^-(\boldsymbol{x}^*) \tag{2}$$

$$\mathsf{Conf}^+(\boldsymbol{x}^*) = \frac{1}{|\widetilde{\mathcal{T}}^+(\boldsymbol{x}^*)|} \sum_{t \in \widetilde{\mathcal{T}}^+(\boldsymbol{x}^*)} \mathsf{c}(t) \quad \text{and} \quad \mathsf{Conf}^-(\boldsymbol{x}^*) = \frac{1}{|\widetilde{\mathcal{T}}^-(\boldsymbol{x}^*)|} \sum_{t \in \widetilde{\mathcal{T}}^-(\boldsymbol{x}^*)} \mathsf{c}(t) \tag{3}$$

where $\widetilde{\mathcal{T}}^+(\boldsymbol{x}^*)$ and $\widetilde{\mathcal{T}}^-(\boldsymbol{x}^*)$ are positive and negative predictions, respectively, and $\lambda$ is a scaling factor that aligns with the ratio of the average standard deviation of two scores across different images. $\mathsf{c}(\cdot)$ denotes the confidence estimate for the prediction; in this paper, we use maximum probability [2].

To separate the positives and negatives from the complete set of predictions, $\widetilde{\mathcal{T}}(\boldsymbol{x}^*)$, we explore three different methods: by (0) considering the **entire** set as positives, (1) applying a **thr**eshold on the maximum probability, (2) selecting the **top-**$k$ predictions, and (3) utilizing non-maximum suppression (**NMS**). For comparison, we also evaluate with the ground truth (**GT**) separation based on DETR's bipartite matching loss with ground truth object annotations. For further details, including how $k$ and the threshold are determined, please refer to Appendices C and D.

## 4.3 Numerical Evaluation

To demonstrate the effectiveness of the proposed method, we compare it extensively with an approach that uses only $\mathsf{Conf}^+$ or $-\mathsf{Conf}^-$. We evaluate using two different DETR models — Deformable-DETR and Cal-DETR — across three datasets — COCO (in-distribution), Cityscapes (near out-of-distribution), and Foggy Cityscapes (out-of-distribution). Empirical results are illustrated in Table 1 above and Table 2 in Appendix D. The key takeaways are as follows.

First, distinguishing between positive and negative queries appears to be crucial for UQ in DETR. Specifically, when confidence scores are averaged without distinguishing positive queries (i.e., $\mathsf{Conf}^+$ w/ entire), the correlation becomes negative. This is because the majority of DETR's outputs are actually negative (i.e., $\mathcal{T}_{\boldsymbol{x}^*} \ll N$), and that confidence scores for negative queries are inversely correlated with $\mathsf{ImReli}$, which is why we report the correlation of "negative(-)" $\mathsf{Conf}^-$.

Second, ContrastiveConf consistently ranks the best or second-best correlation. Although the baselines, $\mathsf{Conf}^+$ and $-\mathsf{Conf}^-$, sometimes surpass ContrastiveConf, their performance fluctuates across settings, highlighting the robustness of the proposed contrastive approach.

Lastly, the performance largely depends on which separation method is applied. For instance, the top-$k$ approach even presents a negative correlation. While the thresholding approach empirically yields better performance, a non-negligible gap remains compared to using ground truth matching. We plan to address this research question, which is discussed further in Appendix E, in a future paper.

---

[2] We empirically observed that the maximum probability, despite its simplicity, often outperforms other estimates, such as negative cross-entropy or layer-across variance [Munir et al., 2024].

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

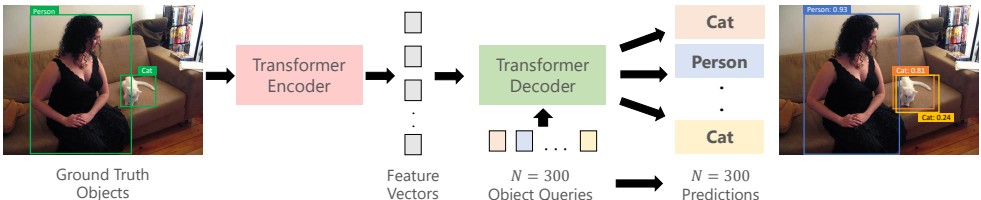

Figure 2: A diagram of the DETR architecture. An input image is first processed through a CNN backbone to generate a 2D feature representation. This representation is then passed to the Transformer encoder, which extracts feature vectors. These feature vectors are sent to the decoder, which receives $N$ learned object queries together. The decoder outputs $N$ prediction sets, each containing a bounding box and corresponding class probabilities. Please refer to the original papers for details.

## A  Bipartite Matching: Positive & Negative Queries

Since the number of queries in DETR, $N$, is much higher than the number of ground truth objects, DETR matches each ground truth object with the corresponding best model prediction during its training. To compute this optimal (i.e., ground truth) matching for the predictions in a given image, a bipartite matching algorithm is applied. More specifically, a matching cost between each pair of a given prediction and a ground truth object is defined as follows:

$$\mathcal{L}_{matching} = \mathcal{L}_{class} + \mathcal{L}_{box} \tag{4}$$

where $\mathcal{L}_{class}$ is the negative prediction confidence of the ground truth class and $\mathcal{L}_{box}$ is the linear combination of the $\ell_1$ loss between the corners of the bounding boxes and $\mathcal{L}_{iou}$. $\mathcal{L}_{iou}$ is the generalized intersection over union (GIoU) loss between bounding boxes. After computing this matching cost for every combination of prediction set and ground truth objects, DETR then efficiently calculates the permutation that minimizes the total matching cost using the Hungarian matching algorithm.

In this paper, we then refer to those matched queries as positive predictions and the remaining unmatched queries as negative predictions. More details on the bipartite matching process and Hungarian algorithm can be found in [Carion et al., 2020].

## B  Further Visualizations

The predictions across all six decoder layers for each of the images presented in the main paper are provided in Figure 3. Generally, the intermediate layers (1, 3, and 4) follow the same trends as those shown previously.

Another noticeable example is provided in Figure 4, where predictions for two different objects, a dog and a laptop, are shown for the same low-reliability image. In Figure 4a, for the laptop object, all of the queries start off with low confidence and remain that way over the course of the layers. However, the class predictions gradually shift from dining table and person to laptop, and result in an overall low confidence score despite a correct final prediction (laptop).

However, Figure 4b indicates that even in a low-reliability image there can still be reliable predictions. For the dog object, the model is very confident in its predictions from the start, with many overlapping prediction sets in the same area and with matching classes. While there are two dominant predictions in the first layer, DETR focuses on a single prediction (query 0) while the confidence for query 1 is gradually reduced; this aligns with the trend observed in the reliable examples shown earlier.

What is demonstrated in Figure 4 not only reveals the limitations of image-level uncertainty quantification but also highlights the need for research on object-level uncertainty. Accordingly, we will

Table 2: Comparison of the proposed method (ContrastiveConf) and baseline methods on their Kendall's $\tau$ correlation coefficient with the image-level reliability in Equation 1, with Deformable-DETR. The GT (oracle), highest, and second-highest scores are highlighted in light gray, blue and green, respectively.

| Method | Separation | CoCo (in-distribution) | | | Cityscapes (near OOD) | | | Foggy Cityscapes (OOD) | | |
|---|---|---|---|---|---|---|---|---|---|---|
| | | Precision | Recall | neg. MC | Precision | Recall | neg. MC | Precision | Recall | neg. MC |
| $Conf^+$ | Entire | -0.417 | -0.429 | -0.397 | -0.203 | -0.219 | -0.257 | -0.131 | -0.138 | -0.092 |
| $Conf^+$ | Thr | 0.346 | 0.293 | 0.581 | 0.240 | 0.209 | 0.362 | 0.224 | 0.178 | 0.276 |
| | Top-$k$ | 0.067 | -0.008 | -0.397 | 0.065 | 0.034 | -0.257 | 0.065 | 0.028 | -0.092 |
| | NMS | -0.402 | -0.417 | -0.382 | -0.192 | -0.212 | -0.244 | -0.122 | -0.131 | -0.083 |
| | GT | 0.447 | 0.351 | 0.799 | 0.472 | 0.450 | 0.776 | 0.487 | 0.471 | 0.790 |
| $-Conf^-$ | Thr | 0.423 | 0.426 | 0.459 | 0.246 | 0.254 | 0.316 | 0.187 | 0.188 | 0.164 |
| | Top-$k$ | 0.296 | 0.275 | 0.442 | 0.179 | 0.207 | 0.289 | 0.132 | 0.138 | 0.145 |
| | NMS | 0.421 | 0.412 | 0.435 | 0.207 | 0.210 | 0.258 | 0.139 | 0.133 | 0.109 |
| | GT | 0.417 | 0.410 | 0.448 | 0.193 | 0.201 | 0.234 | 0.144 | 0.148 | 0.118 |
| ContrastiveConf | Thr | 0.413 | 0.386 | 0.585 | 0.277 | 0.267 | 0.392 | 0.248 | 0.227 | 0.262 |
| | Top-$k$ | 0.342 | 0.294 | -0.091 | 0.233 | 0.224 | -0.028 | 0.167 | 0.144 | 0.082 |
| | NMS | 0.054 | 0.002 | 0.123 | -0.012 | -0.049 | -0.019 | 0.007 | -0.030 | 0.011 |
| | GT | 0.471 | 0.379 | 0.806 | 0.481 | 0.460 | 0.770 | 0.494 | 0.477 | 0.782 |

present a systematic approach for object-level uncertainty quantification and its integration into the image-level problem in our forthcoming work.

## C  Experimental Details

For the `Cityscapes` [Cordts et al., 2016] and `Foggy Cityscapes` [Sakaridis et al., 2018] dataset shown in Table 1 and 2, the classes were mapped to those of the `COCO` [Lin et al., 2014] dataset on which Deformable-DETR and Cal-DETR were trained. Several classes that exist in `Cityscapes` and `Foggy Cityscapes` were left out of this mapping because they are intended for image segmentation and do not translate well to an object detection setting (such as the sky, road, and building classes). Out of the thirty classes in `Cityscapes`, the following eight were used: person, bicycle, car, motorcycle, rider, bus, train, and truck. All of these were directly mapped to their corresponding class in the `COCO` dataset, except for rider which was mapped to person.

For the non-maximum suppression (NMS) results, we use a threshold of $0.65$ on intersection over union (IoU) to determine whether two predictions have sufficient overlap. This is a common choice among related works [Ouyang-Zhang et al., 2022].

For the threshold and top-$k$ methods, we conduct a grid search over $\{0.1, 0.2, \cdots, 0.9\}$ and $\{1, 2, 5, 10, 50, 100, 200\}$, respectively, and reported the best correlation for each configuration. The ablation study on the effect of these parameters is presented in Appendix D.

## D  Further Results

In addition to examining the correlation between different methods and image-level reliability with Cal-DETR, we also present the results using Deformable-DETR in Table 2. Overall, a similar trend can be observed.

Additionally, we perform ablation studies to evaluate the impact of hyperparameters on the correlation, with results illustrated in Figure 5 (threshold) and Figure 6 (top-$k$). As shown, $Conf^+$ achieves high performance when hyperparameters are properly tuned, while $Conf^-$ maintains robust performance. We observe that ContrastiveConf effectively leverages strengths from both $Conf^+$ and $Conf^-$, robustly demonstrating a strong correlation with image-level reliability.

## E  Remaining Future Works: Query-Level UQ

As discussed in previous sections, high-reliability predictions for specific objects can still occur within low-reliability images, highlighting the limitations of image-level UQ. However, the main challenge with query-level (i.e., object-level) UQ in DETR is that *it remains unclear which object in the image each query is attempting to predict*.

Reiterating, previous object-level analysis relies on predefined ground truth objects, which are absent in real-world scenarios. While CNN models benefit from well-established post-processing techniques like NMS, making it easier to obtain positive predictions, DETR and its variants are shown to function differently. DETRs are trained with set loss, resulting in a set of predictions that evolve dynamically through the self-attention mechanism during the decoding process.

This issue is closely tied to the challenge of effectively distinguishing between positive and negative queries within the entire set of predictions in DETR. By effectively separating them, we can not only apply existing object-level UQ methods but also extend them to image-level UQ problems. Currently, we only have preliminary results on this, which are not yet included in this paper, but we are aiming to publish our findings with more comprehensive studies in an upcoming publication.

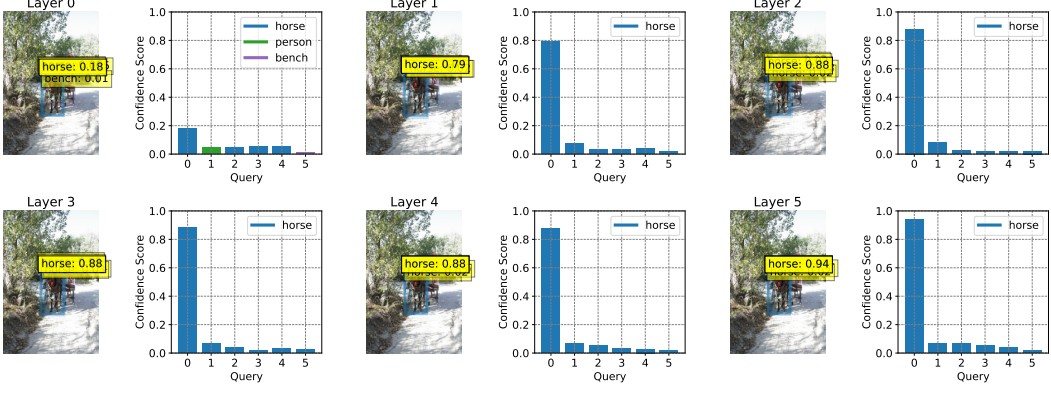

(a) Predictions for a High-Reliability Image: Horse

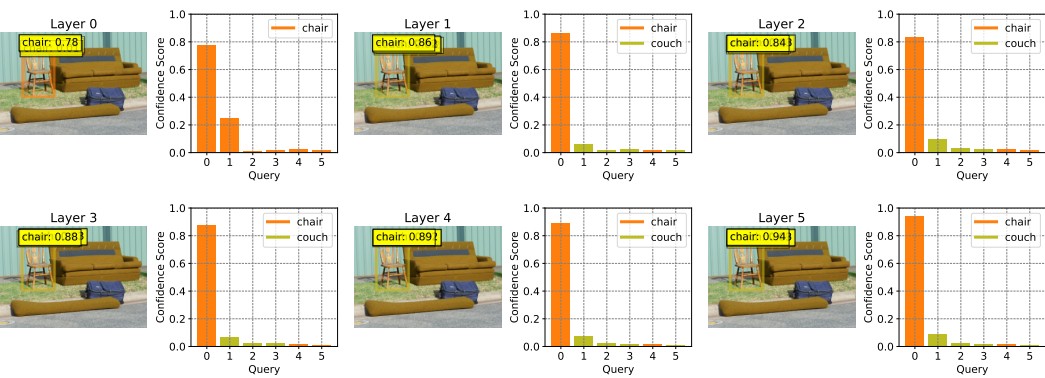

(b) Predictions for a High-Reliability Image: Chair

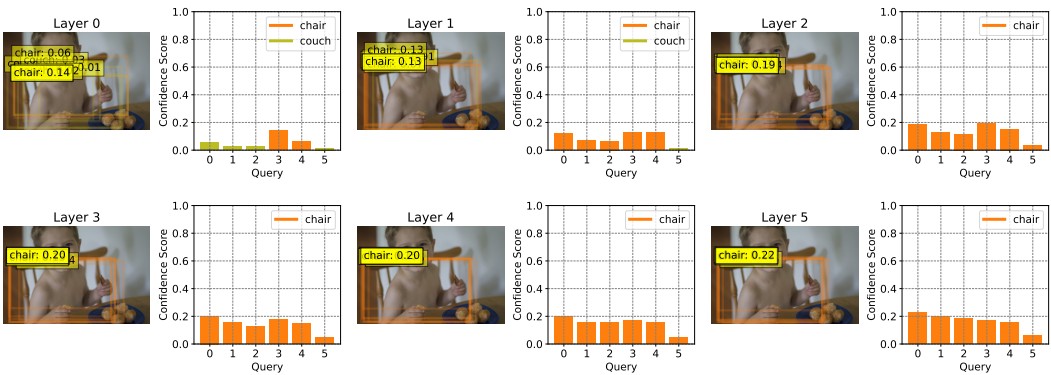

(c) Predictions for a Low-Reliability Image: Chair

Figure 3: Predictions made by Cal-DETR for high- and low-reliability images across all six layers. The positive query and the five negative queries with the largest IoU are presented.

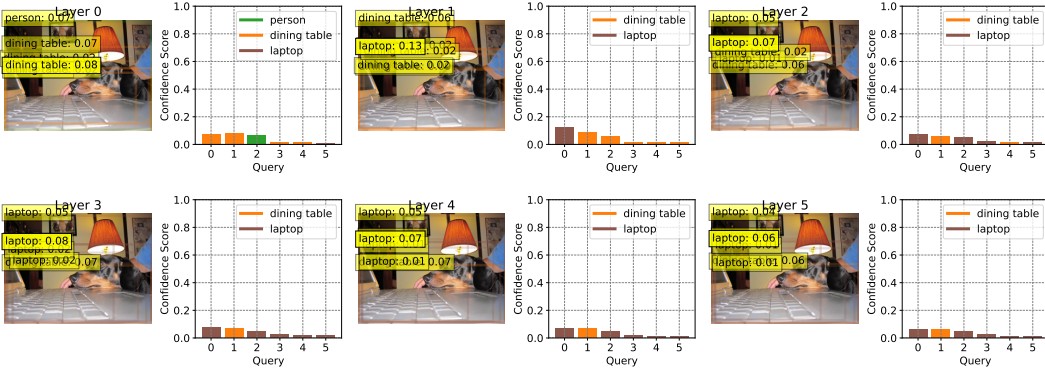

(a) Predictions for a Low-Reliability Object: Laptop

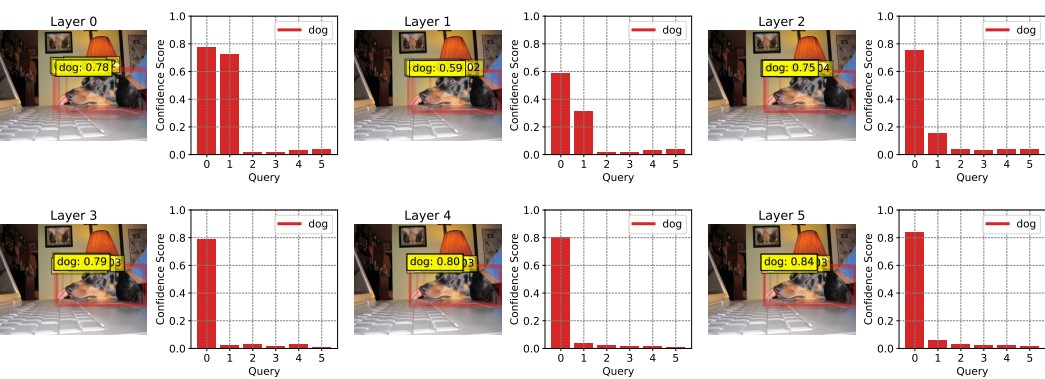

(b) Predictions for a High-Reliability Object: Dog

Figure 4: Predictions made by Cal-DETR for two different objects (one reliable and one unreliable) in a low-reliability image. The positive query and the five negative queries with the largest IoU are presented.

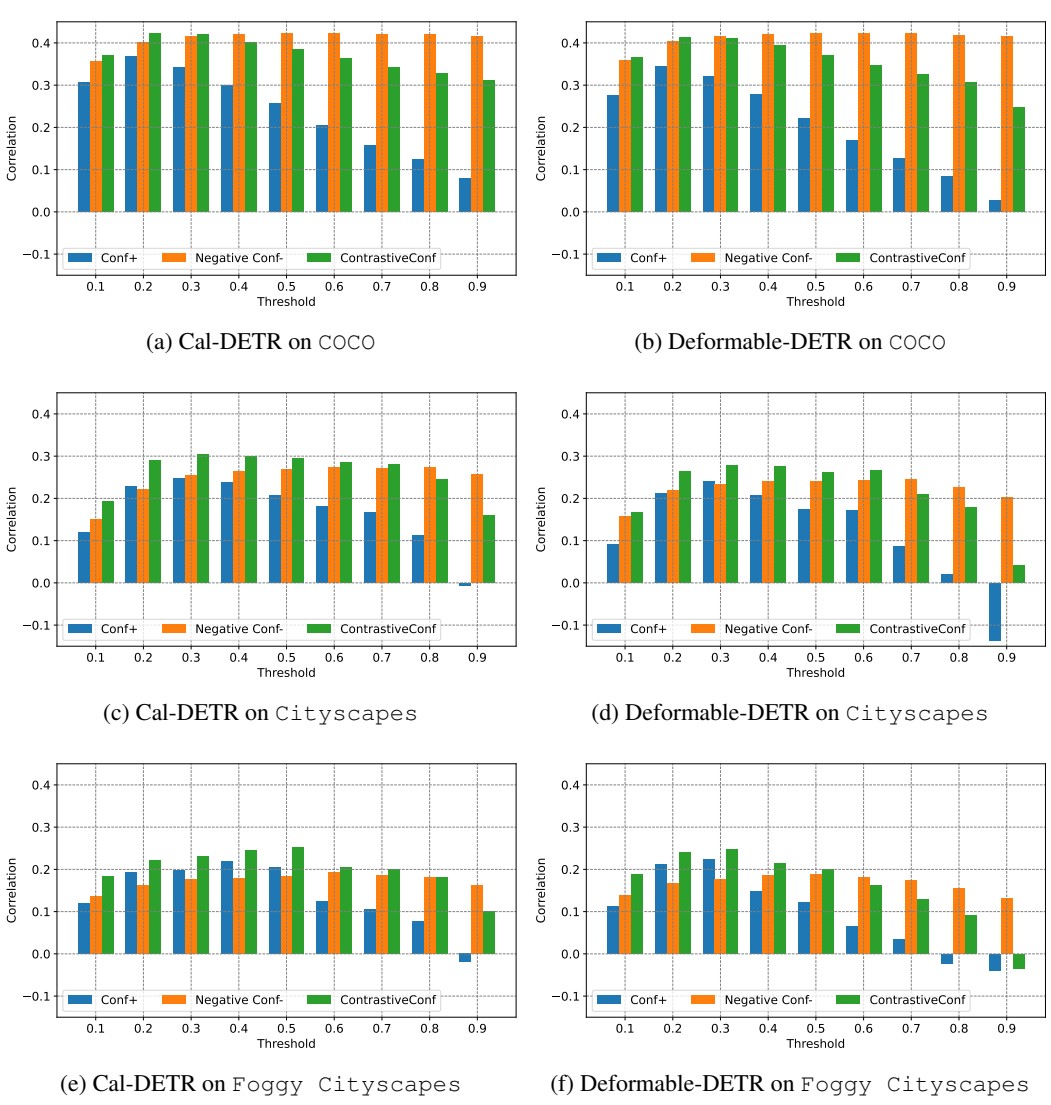

(a) Cal-DETR on `COCO`

(b) Deformable-DETR on `COCO`

(c) Cal-DETR on `Cityscapes`

(d) Deformable-DETR on `Cityscapes`

(e) Cal-DETR on `Foggy Cityscapes`

(f) Deformable-DETR on `Foggy Cityscapes`

Figure 5: Ablation over the threshold used to separate the positive and negative queries.

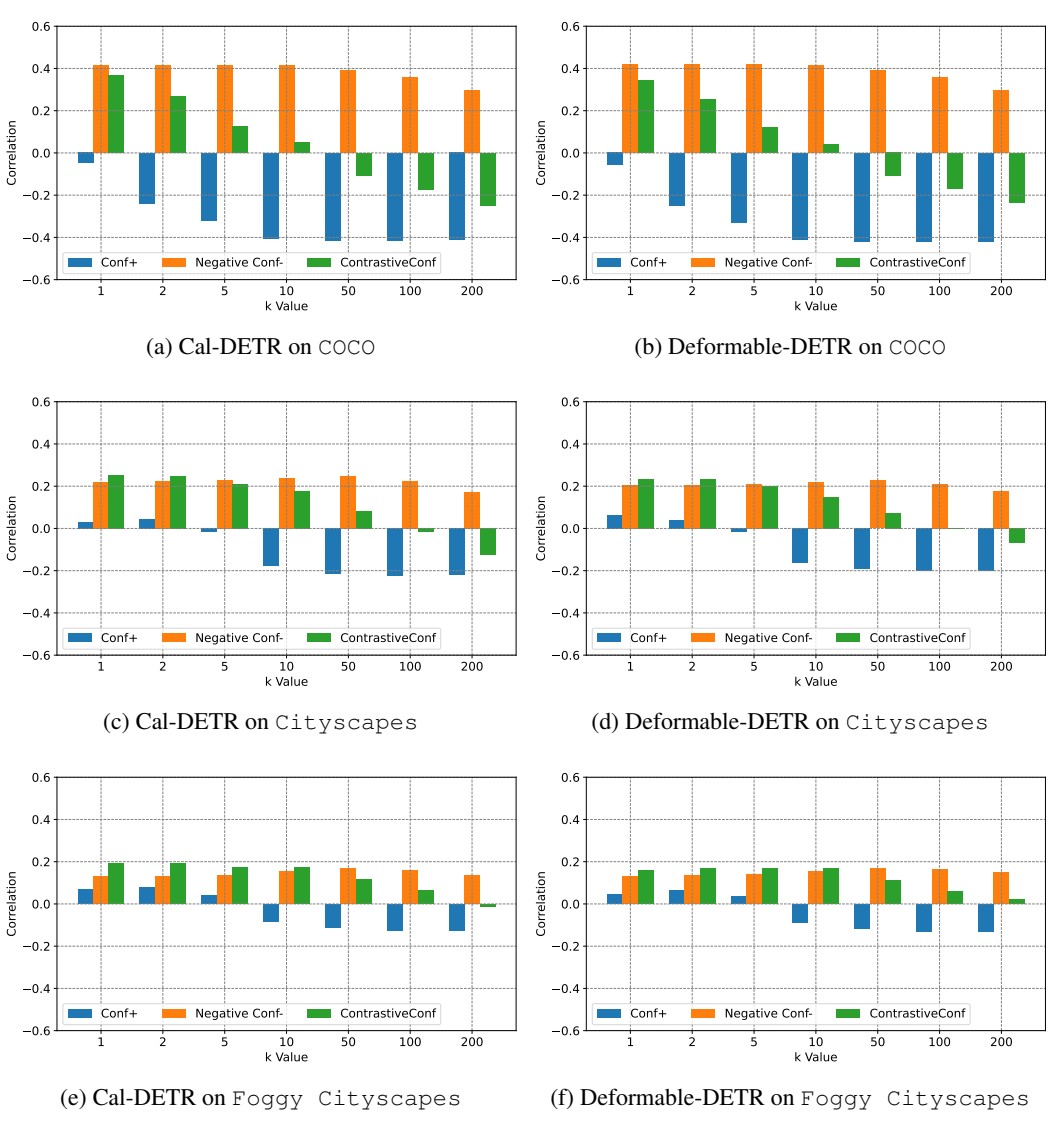

(a) Cal-DETR on COCO

(b) Deformable-DETR on COCO

(c) Cal-DETR on Cityscapes

(d) Deformable-DETR on Cityscapes

(e) Cal-DETR on Foggy Cityscapes

(f) Deformable-DETR on Foggy Cityscapes

Figure 6: Ablation over the $k$-value for the top-$k$ method used to separate the positive and negative queries.

