# OpenReview forum: "Understanding and Quantifying Reliability in Object Detection Transformers"
_NeurIPS.cc/2024/Workshop/BDU — Submitted to NeurIPS BDU Workshop 2024_

### Official Review · Reviewer_qhQf · 2024-09-14
**Evaluation of the Manuscript on Post Hoc Uncertainty Quantification in DETR: Strengths, Limitations, and Areas for Improvement**

**Rating:** 5
**Confidence:** 4

**Review:**

The manuscript developed the importance of distinguishing between these positive and negative predictions from the perspective of post hoc UQ in DETR. The overall quality of the article is general, and the research on the target detection transformer in the research background is insufficient. For example, the overall framework of DETR is divided into several parts, how to play a role, etc. The prediction in the same image has different roles, which is highly original and important for the technical innovation of DETR and the application of complex scenes, but it is still necessary to explore the role of smart prediction from multiple dimensions.

Pros:
1. The manuscript developed the importance of distinguishing between these positive and negative predictions from the perspective of post hoc UQ in DETR. The structure of the article is complete and the logic is clear.

Cons:
1. The overall quality of the article is general. In the introduction, you need to connect the state of the art to your paper goals. Please follow the literature review by a clear and concise state of the art analysis. This should clearly show the knowledge gaps identified and link them to your paper goals. Please reason both the novelty and the relevance of your paper goals. The research on the target detection transformer in the research background is insufficient. For example, the overall framework of DETR is divided into several parts, how to play a role, etc.
2. The prediction in the same image has different roles, which is highly original and important for the technical innovation of DETR and the application of complex scenes, but it is still necessary to explore the role of smart prediction from multiple dimensions.
3. Please revise the conclusion in paragraphs. Conclusions are not just about summarising the key results of the study. It should highlights the insights and the applicability of your findings for the further work. Please make it more concise and show only the high impact outcomes. Report your conclusions in one or maximum 2 paragraph. Avoid bullet form.

---

### Official Review · Reviewer_AGUC · 2024-10-04
**The paper titled "Understanding and Quantifying Reliability in Object Detection Transformers" presents a novel exploration into the reliability of Object Detection Transformers (DETR) models, focusing on uncertainty quantification (UQ). It highlights the importance of differentiating between positive and negative queries and proposes methods for quantifying image-level reliability based on confidence scores. While the research is promising, several areas could benefit from further refinement.**

**Rating:** 4
**Confidence:** 4

**Review:**

Strengths:

Relevant Topic: The focus on UQ in DETR models is timely, addressing a gap in current object detection literature, especially with applications in real-world settings like autonomous driving and medical image analysis.

Novel Contribution:

The differentiation between positive and negative queries to improve UQ is a valuable contribution that adds depth to the understanding of DETR’s internal mechanisms.

Solid Empirical Evaluation:

The paper provides empirical validation across multiple datasets, including COCO and Cityscapes, enhancing the robustness of the proposed methods.


Weaknesses:
Limited Clarity in Methodology: The proposed ContrastiveConf method, while innovative, could benefit from clearer explanations regarding parameter selection and its generalizability to different DETR variants. The scaling factor λ is briefly mentioned but lacks a thorough theoretical justification.
Inadequate Discussion of Failure Cases: Although the authors highlight cases where the model performs poorly (e.g., low-reliability images), more in-depth analysis of these failure cases would strengthen the paper’s conclusions.
Limited Focus on Object-Level UQ: The paper mentions object-level UQ as future work, but further preliminary insights or analysis in this area could enhance the practical utility of the findings, especially for tasks where object-level accuracy is crucial.
Recommendations for Improvement:
Improve Clarity: Clarifying certain methodological details, such as how the threshold for separating positive and negative queries is chosen and why specific parameter values were selected, would make the work more accessible to a broader audience.
Expand the Scope of Experiments: Including more diverse datasets or real-world scenarios would help demonstrate the broader applicability of the proposed methods.
Discuss Scalability: Given that DETR models are often used in real-time or resource-constrained environments, a discussion on the computational cost of the proposed UQ methods would be beneficial.

---

### Decision · Program_Chairs · 2024-10-09

**Decision:**

Reject

**Comment:**

While this paper has promising ideas regarding uncertainty, the reviewers find it not sufficiently ready to be presented due to issues with the writing and presentation. I encourage the authors to improve the work and resubmit to a future workshop.